# Serum FSH as a Useful Marker for the Differential Diagnosis of Ovarian Granulosa Cell Tumors

**DOI:** 10.3390/cancers14184480

**Published:** 2022-09-15

**Authors:** Ayumu Matsuoka, Shinichi Tate, Kyoko Nishikimi, Tastuya Kobayashi, Satoyo Otsuka, Makio Shozu

**Affiliations:** Department of Gynecology, Chiba University Hospital, 1-8-1 Inohana, Chuo-ku, Chiba 260-8677, Japan

**Keywords:** granulosa cell tumor, ovarian tumor, follicle-stimulating hormone, diagnosis

## Abstract

**Simple Summary:**

This study aimed to evaluate whether serum gonadotropin and sex steroid hormone measurement is useful in the differential diagnosis of granulosa cell tumors (GCTs). Serum hormone levels were measured preoperatively in patients who underwent surgery for ovarian tumors (*n* = 471) and compared in two groups, a GCT group (*n* = 13) and a group with other histological types of ovarian tumor (non-GCT) (*n* = 458). Univariate analysis showed that the GCT group had significantly lower serum gonadotropin levels and significantly higher serum sex steroid hormone levels than the non-GCT group. Multivariate analysis revealed that the serum follicle-stimulating hormone (FSH) level was significantly associated with GCT (*p* = 0.004). Receiver-operating characteristic curve analysis for the diagnosis of GCTs showed that the area under the curve of the serum FSH level was 0.99, with a sensitivity of 100% and a specificity of 98%. Preoperative serum FSH levels are an extremely useful marker for differentiating GCTs from all ovarian tumors.

**Abstract:**

Background: We evaluated whether the serum hormone levels are useful in the differential diagnosis of granulosa cell tumors (GCTs), regardless of menopausal status. Methods: Serum levels of luteinizing hormone (LH), follicle-stimulating hormone (FSH), testosterone, estradiol, and progesterone were measured preoperatively in all patients (*n* = 471) who underwent surgery for ovarian tumors at Chiba University Hospital between 2009 and 2021. These were compared in two groups, a GCT group (*n* = 13) and a group with other histological types (non-GCT) (*n* = 458). Results: The GCT group had significantly lower serum LH and FSH (*p* = 0.03 and *p* < 0.001, respectively) and significantly higher testosterone, estradiol, and progesterone (*p* < 0.001, *p* < 0.001, and *p* = 0.045, respectively) than the non-GCT group. Multivariate analysis revealed that serum FSH and estradiol were significantly associated with GCT (FSH, odds ratio (OR) = 0.0046, 95% confidence interval (CI) = 0.0026–0.22, *p* = 0.004; estradiol, OR = 0.98, 95% CI = 0.96–0.998, *p* = 0.046). Receiver-operating characteristic curve analysis for GCTs showed that the area under the curve of serum FSH was 0.99, with a sensitivity of 100% and a specificity of 98%, when the cutoff level was set at 2.0 IU/L. Conclusions: Preoperative serum FSH level is an extremely useful marker for differentiating GCTs from all ovarian tumors.

## 1. Introduction

Granulosa cell tumors (GCTs) are ovarian sex cord stromal tumors that account for 1–2% of all ovarian tumors and 5% of ovarian malignancies [1,2]. It is well documented that GCTs are the most representative hormone-producing ovarian tumors and may occur due to symptoms caused by estradiol from the tumors. GCTs may cause menstrual irregularities and secondary amenorrhea in premenopausal women and genital bleeding in postmenopausal women [3]. Some cases of endometrial hyperplasia and carcinoma have been reported due to hyperestrogenism caused by GCTs [4]. However, approximately 30% of GCTs do not produce estradiol. Therefore, estradiol is not a useful tumor marker for differential diagnosis and follow-up purposes [3,5]. Serum estradiol levels are also increased in half of the epithelial ovarian tumors, mainly ovarian mucinous tumors, and increased serum sex steroid hormone levels have been reported to be useful in estimating mucinous tumor histology after menopause [6,7].

Adult GCTs have traditionally been treated as borderline malignancies. However, since 2014, the World Health Organization’s classification has designated GCTs as malignant neoplasms. Patients with stage I disease have a 5 year survival rate of 75–95% compared with 22–50% in patients with advanced stage (III–IV) [8]. Because the residual disease is significantly associated with prognosis, differentiation of GCTs from benign or borderline malignant ovarian tumors, including other sex cord stromal tumors, is important [9]. Surgery similar to that for epithelial ovarian cancer is recommended for GCT. However, unlike epithelial ovarian cancer, retroperitoneal lymphadenectomy can be omitted [10]. This was based on a meta-analysis of retrospective studies in which lymphadenectomy did not change prognosis and of four retrospective studies in which no positive cases of metastasis were found among 164 patients who underwent lymphadenectomy during initial treatment [11,12,13,14]. Therefore, it is extremely important to distinguish GCTs from other ovarian cancers and to determine the surgical procedure.

GCTs have a variety of gross structures and show different imaging features [15,16]. Because of changes in the pattern and distribution of the tumors associated with hemorrhage or necrosis, GCTs vary in morphology, from completely solid to mixed solid and cystic parts; those with a cystic part have a predominantly and partially solid component. Therefore, imaging findings such as those from CT and MRI are also diverse, making it difficult to differentiate GCTs from epithelial ovarian malignancies [17,18,19,20].

In the present study, we measured preoperative serum gonadotropin and sex steroid hormone levels in all patients who underwent surgery at our hospital. We aimed to investigate whether these serum hormone levels are useful for the preoperative diagnosis of GCTs, regardless of menopausal status.

## 2. Materials and Methods

### 2.1. Patients

All patients who underwent primary surgery for ovarian tumors at the Department of Reproductive Medicine, Graduate School of Medicine, Chiba University, between 2009 and 2021 were included. Postmenopausal women were defined as those who experienced natural menopause for more than 1 year. Patients who were pregnant, those on hormonal therapy, those with central amenorrhea due to pituitary tumors, and those whose preoperative hormone measurements were lost due to emergency hospitalization were excluded. The protocols were approved by the Institutional Review Board (#2124) for Human Research at Chiba University, Chiba, Japan. Patient consent was waived owing to the retrospective design of this study; instead, an opt-out system is used.

### 2.2. Serum Sample and Clinical Data Collection

Serum samples were prospectively collected during the initial visit to our hospital. Serum samples at the time of course change were examined in patients who changed their treatment to surgery during outpatient follow-up. In premenopausal patients, the day of menstruation and phase of menstrual cycle were recorded when the sample was collected. Serum levels of luteinizing hormone (LH; catalog no. 2P40-25), follicle-stimulating hormone (FSH; catalog no. 7K75-25), estradiol (catalog no. 7K72-25), progesterone (catalog no. 7K77-25), and testosterone (catalog no. 2P13-25) were measured using a chemiluminescent immunometric assay (Abbott Japan, Chiba, Japan). Clinical data were collected from patient medical records. Body mass index was calculated as the ratio of weight (kg) to height squared (m^2^). All patients with GCTs were examined for *FOXL2* mutations (c.402CNG; C134W).

### 2.3. Statistical Analyses

Patient characteristics were compared using χ^2^ and Fisher’s exact tests. Serum gonadotropin and sex steroid hormone levels were not normally distributed; thus, the Mann–Whitney test was used to evaluate the differences in serum hormone levels between patients with GCT (GCT group) and those with other histological types of ovarian tumor (non-GCT group). Continuous and categorical variables showing significant differences were analyzed by multivariate logistic regression analysis to screen for independent discriminant features. Receiver operating characteristic (ROC) curve and area under the curve (AUC) analyses were performed to determine the diagnostic value of the serum hormone levels. These statistical analyses were conducted using JMP version 15 (SAS Institute, Cary, NC, USA). A *p*-value less than 0.05 was considered significant.

## 3. Results

### 3.1. Clinical and Pathological Findings

A total of 619 patients underwent surgery for ovarian tumors at our hospital during the relevant period. Among these patients, 67 were pregnant, 21 were on hormone therapy, two had central amenorrhea due to pituitary tumors, and 58 lacked hormone measurement, all of which were excluded from the study. As shown in Figure 1, 471 patients met the inclusion criteria.

The characteristics of all patients are shown in Table 1, and the histological types of ovarian tumors are shown in Appendix A. Of the 471 ovarian tumor patients, 13 (3%) were included in the GCT group, and 458 (97%) were included in the non-GCT group. All the GCT patients were diagnosed as having adult-type GCT through immunohistochemical analysis, and *FOXL2* mutation was confirmed in nine (69%) of 13 GCT patients (Appendix A). Between the two groups, nulliparous women were more common in the GCT group, and preoperative serum cancer antigen 125 and carbohydrate antigen 19-9 levels were significantly higher in the non-GCT group than in the GCT group. There were no significant differences in age, menopausal status, or body mass index between the two groups.

### 3.2. Serum Hormonal Levels in the GCT and Non-GCT Groups

Comparison of serum gonadotropin and sex steroid hormone levels between the two groups showed that the GCT group had significantly lower serum LH and FSH levels and significantly higher and serum testosterone, estradiol, and progesterone levels than the non-GCT group (Figure 2) (Appendix A). Mean preoperative serum hormonal levels in the GCT and non-GCT groups were as follows: LH, 9.3 ± 10.8 and 16.2 ± 12.3 IU/L, respectively (*p* = 0.03); FSH, 0.77 ± 0.70 and 35.3 ± 28.0 IU/L, respectively (*p* < 0.001); testosterone, 1.1 ± 1.1 and 0.51 ± 0.95 nmol/L, respectively (*p* < 0.001); estradiol, 112 ± 107 and 44.5 ± 62.2, respectively (*p* < 0.001); progesterone, 0.93 ± 1.3 and 0.81 ± 1.7 nmol/L, respectively (*p* = 0.045).

In a subgroup analysis of premenopausal patients (*n* = 162), the GCT group (*n* = 7) had a lower serum FSH level and higher serum testosterone level than the non-GCT group (*n* = 155), with no significant differences in other hormonal levels. The serum hormonal levels in the GCT and non-GCT groups of premenopausal patients were as follows: LH, 11.1 ± 11.9 and 6.7 ± 7.8 IU/L, respectively (*p* = 0.49); FSH, 0.80 ± 0.68 and 10.0 ± 12.7 IU/L, respectively (*p* < 0.001); testosterone, 1.4 ± 1.3 and 0.56 ± 1.1 nmol/L, respectively (*p* = 0.0012); estradiol, 106 ± 121 and 85.2 ± 91.2 pmol/L, respectively (*p* < 0.65); progesterone, 1.2 ± 1.5 and 1.7 ± 2.9 nmol/L, respectively (*p* = 0.27). In a subgroup analysis of postmenopausal patients (*n* = 309), the GCT group (*n* = 6) had lower serum LH and FSH levels and higher serum estradiol levels than the non-GCT group (*n* = 303), and the serum testosterone and progesterone levels were not significantly different. The serum hormonal levels in the GCT and non-GCT groups of postmenopausal patients were as follows: LH, 6.8 ± 9.7 and 21.1 ± 11.3 IU/L, respectively (*p* = 0.009); FSH, 0.73 ± 0.81 and 48.4 ± 24.6 IU/L, respectively (*p* < 0.001); testosterone, 0.55 ± 0.31 and 0.49 ± 0.88 nmol/L, respectively (*p* = 0.21); estradiol, 120 ± 94 and 23.9 ± 19.5 pmol/L, respectively (*p* < 0.001); progesterone, 0.40 ± 0.30 and 0.44 ± 0.59 nmol/L, respectively (*p* = 0.65).

The multivariate logistic regression analysis of all patients showed that GCT was significantly associated with serum FSH and estradiol levels (FSH, odds ratio (OR) = 0.0046, 95% confidence interval (CI) = 0.0026–0.22, *p* = 0.004; estradiol, OR = 0.98, 95% CI = 0.96–0.998, *p* = 0.046) (Table 2).

### 3.3. Accuracy of Serum FSH and Estradiol in GCT Diagnosis

An ROC curve analysis of serum FSH and estradiol levels for diagnosing GCTs was performed (Figure 3). The ROC curve in all patients (*n* = 471) showed that the AUC of the serum FSH level was 0.993 (95% CI = 0.981–0.997), with a sensitivity of 100% and a specificity of 98% when the cutoff level was set as 2.0 IU/L, whereas the AUC of the serum estradiol level was 0.81 (95% CI = 0.677–0.893), with a sensitivity of 83% and a specificity of 76% when the cutoff level was 52 pmol/L (Table 2). In ROC curves for premenopausal patients only (*n* = 162), the AUC and cutoff levels, sensitivity, and specificity of serum FSH and estradiol levels were as follows: FSH, AUC = 0.979, 95% CI = 0.944–0.993, cutoff level = 1.8 IU/L, sensitivity = 95%, specificity = 95%; estradiol, AUC = 0.550, 95% CI = 0.315–0.765, cutoff level = 64 pmol/L, sensitivity = 71%, specificity = 56%. Subsequently, the ROC curves for postmenopausal patients only (*n* = 309) showed that the AUC and cutoff levels, sensitivity, and specificity of serum FSH and estradiol levels were as follows: FSH, AUC = 1.00, cutoff level = 2.0 IU/L, sensitivity = 100%, specificity = 100%; estradiol, AUC = 0.960, 95% CI = 0.891–0.986, cutoff level = 52 pmol/L, sensitivity = 100%, specificity = 91%.

The ROC curve between GCTs (*n* = 13) and malignant tumors (*n* = 278) showed that the AUCs of serum FSH and E2 level were 0.994 (95% CI = 0.983–0.998) and 0.832 (95% CI = 0.702–0.912), respectively. Between GCTs (*n* = 13) and borderline tumors (*n* = 48), the AUCs of serum FSH and E2 levels were 1.00 and 0.752 (95% CI = 0.624–0.843), respectively. Subsequently, the ROC curves between GCTs (*n* = 13) and benign tumors (*n* = 132) showed that the AUCs of serum FSH and estradiol levels were 0.981 (95% CI = 0.963–0.998) and 0.782 (95% CI = 0.682–0.876), respectively.

## 4. Discussion

### 4.1. Key Findings of This Study

GCTs are representative hormone-producing ovarian tumors. In this study, we measured the preoperative serum gonadotropin and sex steroid hormone levels in all patients who underwent surgery for ovarian tumors at our hospital regardless of menopausal status. Our study revealed that the AUC of serum FSH level for preoperative diagnosis of GCTs from other ovarian tumors was 0.99. When the cutoff level for serum FSH was set at 2.0 IU/L, the sensitivity and specificity were 100% and 98%, respectively. GCTs are difficult to differentiate from epithelial ovarian tumors using preoperative CT and MRI images [19]. Unlike epithelial ovarian cancer, retroperitoneal lymphadenectomy is not recommended in initial staging surgery of GCTs due to the extremely low lymph node metastasis rate. Preoperative estimation of GCTs by measuring serum FSH levels can help determine the patient treatment strategies and surgical procedures.

### 4.2. Relationship between the Serum Hormone Levels and GCTs

In the present study, we compared the preoperative serum hormone levels in GCTs with those in other ovarian tumors. We found that GCTs had significantly lower serum gonadotropin and higher sex steroid hormone levels. In the multivariate analysis, serum FSH and estradiol levels were independently associated with GCTs. Inhibin is secreted by ovarian granulosa cells and acts on the anterior pituitary to suppress FSH secretion and regulate the number of developing follicles [21]. Excessive increase of inhibin leads to a marked decrease in FSH, resulting in decreased estrogen secretion in normal ovaries. Inhibin consists of an α subunit linked to either a βA or a βB subunit, forming inhibin A and B, respectively. Most GCTs produce inhibin, and 94% of these tumors are immunohistologically positive for inhibin [22]. Inhibin B increased in 89% and inhibin A increased in 67% of patients with GCTs [23]. Both inhibin A and B produced by GCTs are involved in a decrease in serum FSH levels by negative feedback [24,25]. Hormonal findings in GCTs that secrete inhibin have been noted to be similar to those in isolated FSH deficiency [26]. In premenopausal women with GCTs, secondary amenorrhea and infertility may occur due to increased levels of inhibin, which subsequently inhibits the secretion of pituitary FSH. Women who complained of infertility have been reported to become pregnant soon after tumor resection. Of the 13 GCT cases in the present study, two patients who had secondary amenorrhea and infertility became pregnant after tumor resection and delivered [27]. In contrast, estradiol is produced in only 70% of GCTs [3,5]. Theca cells may be involved in this estrogen-producing capacity of GCTs. GCT cases without increased levels of serum estradiol are suspected to lack theca cells, which produce androstenedione, a necessary precursor for estradiol synthesis. Kitamura et al. showed that theca cells have P450α-hydroxylase (P450c17) expression that was significantly associated with elevated serum estradiol levels in GCT patients [5]. Subgroup analyses of premenopausal patients in this study also showed that serum estradiol levels were not significantly different between GCTs and other histological types. To date, there is no established evidence for LH change in GCT patients. We speculate that progesterone feedback may be related to low LH in GCT patients in this study. Serum testosterone levels of premenopausal patients were higher in those with GCTs. Previous cases of testosterone-producing GCTs have also been reported [28,29]. FSH receptors are present in granulosa cells, and FSH induces the conversion of androgens to estrogen in granulosa cells. In GCTs in which FSH is suppressed, this aromatization may be suppressed, resulting in increased serum testosterone levels.

### 4.3. Diagnostic Value of the Serum FSH Level for GCTs

Measurement of serum FSH levels was helpful for preoperative diagnosis of GCT, and the sensitivity and specificity of the cutoff level of 2.0 IU/L were both favorable. In our institution, serum hormone levels, such as LH, FSH, testosterone, estradiol, and progesterone, which are commonly used to assess the hormonal status of women, are measured before surgery for ovarian tumors in all patients. To date, serum inhibin level has been reported to be useful in the differential diagnosis of GCTs [30,31]. Haltia et al. evaluated the usefulness of serum inhibin B level in the diagnosis of GCTs in the control group of epithelial ovarian cancer and endometriotic cyst [30]. However, inhibin is also believed to be produced in 70–80% of cases of ovarian mucinous neoplasia [24,25,32]. The report by Haltia et al. was limited by the small number of patients with potentially inhibin-producing mucinous tumors because only two patients with mucinous ovarian carcinoma were included in the control group. Since GCTs are most often multilocular and solid on imaging, differentiation from ovarian mucinous carcinoma and mucinous borderline malignancy is important [20]. In our study, 458 patients included 19 with mucinous carcinomas, 29 with mucinous borderline malignancies, and 40 with mucinous cystadenomas. The mean serum FSH levels for these mucinous neoplasms were 27.1 ± 2.9 IU/L for the total patients (*n* = 88), 10.1 ± 4.0 IU/L for premenopausal patients (*n* = 21), and 24.8 ± 3.0 IU/L for postmenopausal patients (*n* = 67) patients. The overall minimum level was 2.4 IU/L for the premenopausal patients. Unlike GCTs, there was no significant decrease in serum FSH levels in patients with mucinous neoplasms. It has been previously reported that FSH is not inhibited by increased serum inhibin in patients with mucinous neoplasms [24,25,32]. It is suggested that inhibin secreted by a mucinous tumor may have little or no biological activity. In premenopausal women, serum FSH levels fluctuate depending on the menstrual cycle. Of the nine false-positive cases with serum FSH levels ≤ 2.0 ng/dL in this study, five were premenopausal patients whose serum samples were collected during their luteal phase. Their serum FSH levels were within the normal range of the luteal phase, and their serum progesterone levels were high. Meanwhile, all patients with GCTs were measured in the secondary amenorrhea or follicular phase. In the differential diagnosis of GCTs for premenopausal women, measuring serum FSH levels during the follicular phase and setting a cutoff level to 1.8 IU/L may further improve diagnostic accuracy.

### 4.4. Role of FSH as a Tumor Marker to Follow up GCTs

Several studies have also examined tumor markers for follow-up after surgery for GCTs. All GCT patients in this study were confirmed to have normal FSH levels after tumor removal. However, the usefulness of serum FSH levels in the diagnosis of the recurrence could not be examined because all of these patients had no recurrence. Serum inhibin and anti-Müllerian hormone (AMH) levels decrease rapidly after surgery in GCTs, becoming extremely low or undetectable [33,34,35]. Recurrence is more likely if serum inhibin levels, once reduced by tumor resection, continue to rise, and increased inhibin levels predict recurrence earlier than clinical manifestations [23,36]. Inhibin levels are also useful in evaluating the response to chemotherapy. Long et al. investigated the usefulness of serum AMH levels in postoperative follow-up of GCTs [37]. Among the 24 patients after surgery, blood AMH levels became undetectable in 23 patients, and only one patient showed an increase in AMH for which there was no evidence of recurrence. In addition, an increase in serum AMH levels was observed in 15 of the 16 recurrent cases, and the remaining case did not show an increase in AMH after recurrence. Recurrent cases of GCTs have a poor prognosis, with an overall mortality of 73% and overall survival of 5.6 years [15]. Although following up on these tumor markers is important for early diagnosis and treatment of recurrence, the usefulness of serum FSH levels for follow-up needs further investigation.

### 4.5. Strengths and Limitations

A limitation of this study was the small sample size of GCT patients. We conducted a retrospective review of all patients who underwent surgery for ovarian tumors at a single institution during a 13 year period. GCTs accounted for 13 of the 471 total ovarian tumors, representing 3% of the total. This was nearly equivalent to the typical incidence rate of GCTs in all ovarian tumors. This study, which had a large number (458) of controls with all histological types, was close to the clinical practice of differentiating GCTs from various histological types. Because GCTs exhibit a variety of morphologies, it is important to differentiate GCTs from all ovarian tumors, both benign and malignant. Secondly, the measurement of serum inhibin level was not performed in this study. In Japan, the measurement of inhibin is not covered by insurance, making it difficult to implement in clinical practice. In addition, inhibin is also complicated because of the types to be measured and methods of measurement [31]. In contrast, serum FSH level, which is suppressed by both inhibin A and B, is commonly used to investigate menstrual disorders. FSH can be measured more easily than inhibin, and its application in clinical practice for the diagnosis of ovarian tumors is expected. Thirdly, a low frequency of *FOXL2* mutations in the granulosa cell tumors was observed in this study. The FOXL2 gene contributes to the development of GCTs through the C134W mutation by promoting cell proliferation through activin, inhibiting apoptosis through apoptosis-related receptors, and promoting estrogen production through the promotion of CYP19 transcription [38]. Previous studies have reported the presence of somatic *FOXL2* mutation in granulosa cell tumors in 94–97% of cases [39,40]. Conversely, the frequency of these mutations has been reported to be approximately 70% in several other studies, as in the present one [41,42,43]. The lower frequency is reportedly related to the use of formalin-fixed and paraffin-embedded (FFPE) specimens and potential tumor heterogeneity. In the present study, nine of the 10 cases analyzed via frozen tissue had *FOXL2* mutation; one of the three cases analyzed using FFPE revealed the mutation.

## 5. Conclusions

Serum FSH level is a useful marker for the preoperative differential diagnosis of GCTs. Regardless of menopausal status, a cutoff level of 2.0 IU/L for serum FSH had a good AUC of 0.99, which was even better for postmenopausal patients. For premenopausal patients, setting the cutoff level of serum FSH at 1.8 IU/L and measuring serum FSH during the follicular phase could improve diagnostic accuracy. It is meaningful in the clinical practice of gynecologic oncology to distinguish GCTs, showing a variety of presentations from all ovarian tumors by simple measurement of serum FSH level. Further investigation with a large sample size is necessary to confirm these findings.

## Figures and Tables

**Figure 1 cancers-14-04480-f001:**
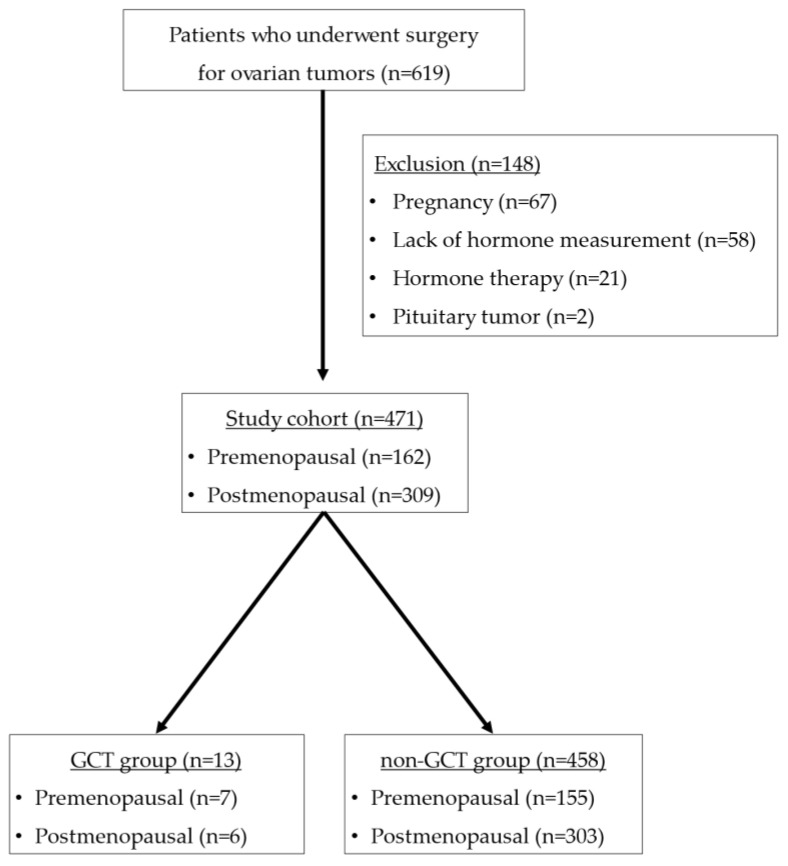
Inclusion and exclusion criteria to identify the study cohort.

**Figure 2 cancers-14-04480-f002:**
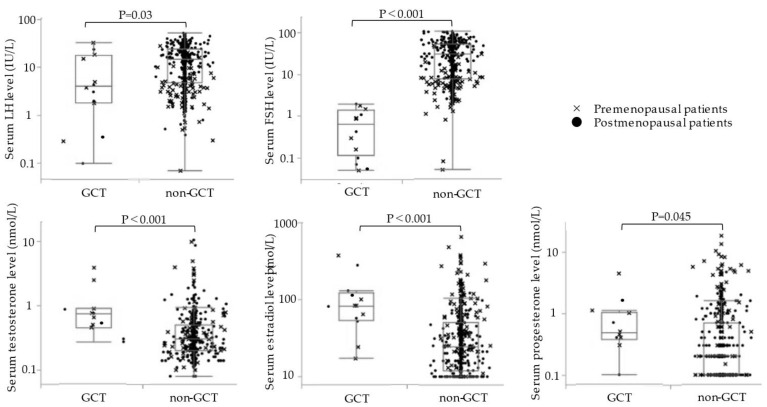
Univariate analyses of serum hormone levels in the GCT (*n* = 13) and non-GCT groups (*n* = 458). Differences between the GCT and non-GCT groups were analyzed for unpaired data using the Mann–Whitney U test. GCT, granulosa cell tumor; FSH, follicle-stimulating hormone; LH, luteinizing hormone.

**Figure 3 cancers-14-04480-f003:**
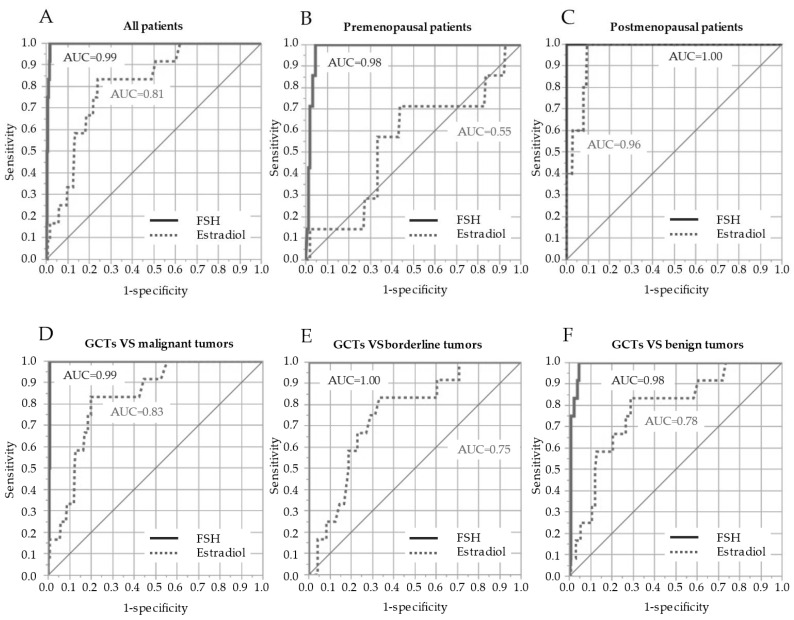
ROC of serum FSH and estradiol levels for the diagnosis of GCTs in all (**A**), premenopausal (**B**), and postmenopausal patients (**C**). ROC between GCTs and malignant tumors (**D**), GCTs and borderline tumors (**E**), and GCTs and benign tumors (**F**) according to malignancy. ROC, receiver operating curves; AUC, area under the curve GCT, granulosa cell tumor; FSH, follicle-stimulating hormone.

**Table 1 cancers-14-04480-t001:** Patient characteristics.

Characteristic	All (*n* = 471)	GCT Group (*n* = 13)	Non-GCT Group (*n* = 458)	*p*-Value
Age (years)	57 (8–92)	44 (23–83)	58 (8–92)	0.08
Menopausal status				
Premenopausal	162 (34)	7 (54)	155 (34)	0.1
Postmenopausal	309 (66)	6 (46)	303 (66)	
Menopause age (years)	50 (40–60)	51 (43–56)	50 (40–60)	0.6
Time from menopause (years)	14 (2–39)	26 (3–33)	14 (2–39)	0.4
Parity				
Nullipara	152 (32)	8 (62)	144 (31)	0.03
Multipara	319 (68)	5 (38)	314 (69)	
BMI (kg/m^2^)	22 (14–42)	21 (16–35)	22 (14–42)	0.1
Preoperative CA125 (U/mL)	83 (5–67,000)	17 (6–240)	90 (5–67,000)	0.005
Preoperative CA19-9 (U/mL)	20 (0.1–75,000)	10 (4–41)	21 (0.1–75,000)	0.03

Values are presented as the median (range) or *n* (%). BMI, body mass index; GCT, granulosa cell tumor; CA125, cancer antigen 125; CA19-9, carbohydrate antigen 19-9.

**Table 2 cancers-14-04480-t002:** Multivariate logistic regression analysis evaluating the associations between serum hormone levels and granulosa cell tumors with receiver operating characteristic curve analysis for granulosa cell tumor.

Features	Multivariate Logistic Regression Analysis			Receiver Operating Characteristic Analysis		
	OR	95% CI	*p*-Value	AUC (95% CI)	Specificity	Sensitivity
FSH	0.046	0.0026–0.22	0.004	0.99 (0.981–0.997)	100	98
Testosterone	0.44	0.12–1.4	0.2			
Estradiol	0.98	0.96–0.998	0.046	0.81 (0.677–0.893)	83	76

OR, odds ratio; CI, confidence interval; AUC, area under the receiver operating characteristic curve; FSH, follicle-stimulating hormone.

## Data Availability

All datasets analyzed during the current study are available from the corresponding author upon reasonable request.

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
