# Peer review of "Serum FSH as a Useful Marker for the Differential Diagnosis of Ovarian Granulosa Cell Tumors"

_cancers, 2022, doi:10.3390/cancers14184480_

Round 1
Reviewer 1 Report
This paper using solid data illustrates that level of follicle-stimulating hormone can be used as a specific biomarker for GCT from other ovarian cancers.
As the author discussed in this paper, the number of GCTs patients is extremely low, due to the lower incidence. My question here is why the finding of the biomarker for GCT is significant in ovarian cancer and its treatment. Please provide more background and evidence to support your work.
Please provide the full name for abbreviations, for example, luteinizing hormone (LH).
Author Response
This paper using solid data illustrates that level of follicle-stimulating hormone can be used as a specific biomarker for GCT from other ovarian cancers.
As the author discussed in this paper, the number of GCTs patients is extremely low, due to the lower incidence. My question here is why the finding of the biomarker for GCT is significant in ovarian cancer and its treatment. Please provide more background and evidence to support your work.
Response: Thank you for raising these important issues. The previous manuscript did not fully describe the background and significance of differentiating GCT from ovarian cancer. We added the following to the Introduction and Discussion sections to highlight the importance of the GCT biomarker. We also added the references that support them.
Adult GCTs have traditionally been treated as borderline malignancies. However, since 2014, the World Health Organization’s classification has designated GCTs as malignant neoplasms. Patients with stage I disease have a 5-year survival rate of 75%–95% compared with 22%–50% in patients with advanced stage (III–IV). Because the residual disease is significantly associated with prognosis, differentiation of GCTs from benign or borderline malignant ovarian tumors, including other sex cord-stromal tumors, is important [8]. Surgery similar to that for epithelial ovarian cancer is recommended for GCT. However, unlike epithelial ovarian cancer, retroperitoneal lymphadenectomy can be omitted [9]. This was based on a meta-analysis of retrospective studies in which lymphadenectomy did not change prognosis and on four retrospective studies in which no positive cases of metastasis were found among 164 patients who underwent lymphadenectomy during initial treatment [10–13]. Therefore, it is extremely important to distinguish GCTs from other ovarian cancers and to determine the surgical procedure.
Please provide the full name for abbreviations, for example, luteinizing hormone (LH).
Response: Thank you. We have added the full names of abbreviations.

Reviewer 2 Report
This is an interesting piece of work by Matsuoka et. al., that has been collected over a long span of time period. Here, it has been shown that FSH levels can be a marker that can distinguish GCTs from all ovarian tumors. There are various undetermined parameters that make the manuscript weak in scientific soundness.
Major comments:
1. What is the aggressiveness of GCTs? Please mention including its 5-year survival rate.
2. Theca cells produce LH. So, what is the role played by theca cells in GCTs, please discuss.
3. AMH and inhibin inhibit FSH during different phases of follicle development. Please explain why there is low FSH and LH. Has the author checked the levels of inhibin or AMH?
4. Please show the data for immunohistochemical analysis and the mutation of FOXL2. Also, mention the genes that are regulated by FOXL2
5. Since FSH and LH levels are low, how is the reproductive health of GCTs patients affected? Please explain in the discussion. Has the author checked the reproductive health for this study?
Minor comments:
1. Please use the same font in the abstract.
2. Please make a table showing pre and post-menopausal patients with GCT, showing differences with pre and post-menopausal patients with non-GCT in serum levels of LH, FSH, estradiol, progesterone, and testosterone.
Author Response
This is an interesting piece of work by Matsuoka et. al., that has been collected over a long span of time period. Here, it has been shown that FSH levels can be a marker that can distinguish GCTs from all ovarian tumors. There are various undetermined parameters that make the manuscript weak in scientific soundness.
Major comments:
- What is the aggressiveness of GCTs? Please mention including its 5-year survival rate.
Response: Thank you for your feedback. GCTs have been classified as a malignant neoplasm according to the 2014 World Health Organization classification. Five-year survival and prognostic factors have been added to the Introduction section. The usefulness of differentiating GCTs from other ovarian tumors was also added to the Introduction and Discussion section.
- Theca cells produce LH. So, what is the role played by theca cells in GCTs, please discuss.
Response: Thank you for raising this important point. GCTs are known for estrogen production, but approximately 30% of GCT cases do not produce estradiol. Theca cells may be involved in this estrogen-producing capacity of GCTs. GCT cases without increased levels of serum estradiol are suspected to lack theca cells, which produce androstenedione, a necessary precursor for estradiol synthesis. Kitamura et al. showed that theca cells have P450α-hydroxylase (P450c17) expression that was significantly associated with elevated serum estradiol levels in GCT patients [5]. We added these descriptions to the Discussion section.
- AMH and inhibin inhibit FSH during different phases of follicle development. Please explain why there is low FSH and LH. Has the author checked the levels of inhibin or AMH?
Response: We did not measure serum AMH or inhibin during the preoperative diagnosis of GCT patients because they are not covered by insurance in Japan. In addition, retrospective measurement of AMH and inhibin was not possible because the serum of more than half of the 13 GCT patients in this study was not stored.
As described in the Discussion section, both inhibin A and B produced by GCTs are involved in the decrease of serum FSH levels by negative feedback. To date, there is no established evidence for LH change in GCT patients. We speculate that progesterone feedback may be related to low LH in GCT patients in this study. We added these descriptions to the Discussion section.
- Please show the data for immunohistochemical analysis and the mutation of FOXL2. Also, mention the genes that are regulated by FOXL2
Response: Thank you for pointing that out. In Section 3.1, Clinical and Pathological Findings, we noted that the GCT cases were diagnosed using immunostaining, which is not immunostaining for FOXL2, but rather general immunostaining to distinguish epithelial ovarian cancers. We apologize for the insufficient description. We did not investigate the FOXL2 immunostaining in our GCT cases, but we examined tumor tissues from all 13 cases for FOXL2 mutations. We revised the description in the manuscript to indicate the direct sequence analyses of cases with and without FOXL2 mutations. We also added a description of FOXL2 and its mechanism as follows.
The FOXL2 gene contributes to the development of GCTs through the C134W mutation by promoting cell proliferation through activin, inhibiting apoptosis through apoptosis-related receptors, and promoting estrogen production through the promotion of CYP19 transcription.
- Since FSH and LH levels are low, how is the reproductive health of GCTs patients affected? Please explain in the discussion. Has the author checked the reproductive health for this study?
Response: Thank you for your comments and questions. In premenopausal women with GCTs, secondary amenorrhea and infertility may occur due to increased levels of inhibin, which subsequently inhibits the secretion of pituitary FSH. Women who complained of infertility have been reported to become pregnant soon after tumor resection. Of the 13 GCT cases in the present study, 2 patients who had secondary amenorrhea and infertility became pregnant after tumor resection and delivered. We described the association between infertility and GCTs and noted that early diagnosis and treatment may preserve fertility, especially in younger GCT patients.
Minor comments:
- Please use the same font in the abstract.
Response: We apologize for using a different font. We have changed the font in the abstract.
- Please make a table showing pre and post-menopausal patients with GCT, showing differences with pre and post-menopausal patients with non-GCT in serum levels of LH, FSH, estradiol, progesterone, and testosterone
Response: Thank you for your suggestion. We added Supplementary Table S1, which presents the serum hormone levels of premenopausal and postmenopausal patients in the GCT and non-GCT groups, respectively.

Round 2
Reviewer 2 Report
I thank the authors for diligently answering all the raised queries.